# A Preliminary Study on the Characteristics of microRNAs in Ovarian Stroma and Follicles of Chuanzhong Black Goat during Estrus

**DOI:** 10.3390/genes11090970

**Published:** 2020-08-21

**Authors:** Tingting Lu, Xian Zou, Guangbin Liu, Ming Deng, Baoli Sun, Yongqing Guo, Dewu Liu, Yaokun Li

**Affiliations:** 1College of Animal Science, South China Agricultural University, Guangzhou 510642, China; 18819255207@163.com (T.L.); zouxian08@163.com (X.Z.); gbliu@scau.edu.cn (G.L.); dengming@scau.edu.cn (M.D.); baolisun@scau.edu.cn (B.S.); yongqing@scau.edu.cn (Y.G.); dwliu@scau.edu.cn (D.L.); 2State Key Laboratory of Livestock and Poultry Breeding, Guangdong Key Laboratory of Animal Breeding and Nutrition, Institute of Animal Science, Guangdong Academy of Agricultural Sciences, Guangzhou 510640, China

**Keywords:** microRNA, ovarian stroma, follicles, follicular maturation, Chuanzhong black goat

## Abstract

microRNAs (miRNAs) play a significant role in ovarian follicular maturity, but miRNA expression patterns in ovarian stroma (OS), large follicles (LF), and small follicles (SF) have been rarely explored. We herein aimed to identify miRNAs, their target genes and signaling pathways, as well as their interaction networks in OS, LF, and SF of Chuanzhong black goats at the estrus phase using small RNA-sequencing. We found that the miRNA expression profiles of LF and SF were more similar than those of OS—32, 16, and 29 differentially expressed miRNAs were identified in OS vs. LF, OS vs. SF, and LF vs. SF, respectively. Analyses of functional enrichment and the miRNA-targeted gene interaction network suggested that miR-182 (*SMC3*), miR-122 (*SGO1*), and miR-206 (*AURKA*) were involved in ovarian organogenesis and hormone secretion by oocyte meiosis. Furthermore, miR-202-5p (*EREG*) and miR-485-3p (*FLT3*) were involved in follicular maturation through the MAPK signaling pathway, and miR-2404 (*BMP7* and *CDKN1C*) played a key role in follicular development through the TGF-β signaling pathway and cell cycle; nevertheless, further research is warranted. To our knowledge, this is the first study to investigate miRNA expression patterns in OS, LF, and SF of Chuanzhong black goats during estrus. Our findings provide a theoretical basis to elucidate the role of miRNAs in follicular maturation. These key miRNAs might provide candidate biomarkers for the diagnosis of follicular maturation and will assist in developing new therapeutic targets for female goat infertility.

## 1. Introduction

Fecundity is a complex quantitative character, which is mainly restricted by genetic factors and has an economic importance in the goat industry [1]. Kidding rate is determined by ovulation rate, which is associated with the reproductive performance of female goats and closely related to follicular development [2]. Follicular development is a very complex process. More than 99% of the follicles in mammalian ovaries undergo atresia, and only a few actually ovulate during follicular development [3,4]. As a dynamic reproductive organ, the ovaries contain ovarian stroma and follicles at different stages of development (e.g., primary follicles, growth follicles, and mature follicles) [5,6]. Because of the complexity of the ovary, a direct study using a whole ovary cannot completely reflect its overall function. Therefore, it is necessary to subdivide the ovary before conducting further research.

microRNAs (miRNAs), a type of non-coding small RNA with a length of 19–24 nucleotides (nt), regulate gene expression at the post-transcriptional level by interfering with or degrading mRNA expression [7,8]. For the regulation of female reproduction, they are mainly involved in the formation of primordial follicles, recruitment and selection of follicles, follicular atresia, ovulation, and luteinization [9,10,11]. Several studies have reported the expression of different miRNAs in the whole ovary and single follicle. For example, miR-145 [12] and miR-143 [13] have been reported to regulate the formation and development of primordial follicles, playing a pivotal role in early follicular formation. The let-7 family, miR-23-27-24 cluster, miR-183-96-182 cluster, and miR-17-92 cluster evidently affect granulosa cell apoptosis and follicular atresia [14]. Further, miR-21 [15] and miR-224 [16] have been found to promote the proliferation of granulosa cells and affect the development of follicles in mice. As evidence from the literature, the miRNA expression profiles of follicles in pigs, cattle, and sheep at different developmental stages show distinct variations. In addition, studies have found that the imbalance of miR-423, miR-33b, and miR-142 may lead to polycystic ovary syndrome [17]. The deletion of the miR-17-92 cluster [18] and miR-7a2 [19] can lead to infertility in female mice. This suggests that miRNA might also affect the reproductive performance of female goats, thus affecting the economic effect of the goat industry. However, the research of goat miRNA mainly focused on the ovaries [20,21], and it is impossible to accurately determine the functional site of miRNAs. Besides, as yet, miRNA expression patterns in ovarian stroma and follicles of goats during estrus have not been reported.

Chuanzhong (CZ) black goat (*Capra hircus*) is an excellent local goat breed in China [22]. A one-week-old goat weighs approximately 20 kg, whereas an adult female goat usually weighs around 50 kg. Sexual maturity is achieved by 3 months, first mating occurs at the age of 5–6 months, and the gestation period is 146–153 days [23]. Upon long-term natural and artificial selection, its reproductive performance has been found to be outstanding, with the genetic performance being stable; the average kidding rate was 236.78%, and the survival rate of the lambs was 91% [24]. Generally, CZ black goat has good reproductive performance, which is an ideal model for studying goat reproductive traits. As one of the important breeding materials in China, CZ black goat can accelerate the breeding process of high-yielding ewes, reduce the breeding cost, and benefit the large-scale and intensive development of the goat industry [25,26,27]. After crossing with local goats, the fecundity and growth performance of their offspring have been improved [28,29]. However, only a few studies have assessed the reproductive performance of CZ black goats.

In this study, we used high-throughput sequencing to analyze the small RNAs of ovarian stroma (OS), large follicles (LF, diameter >10 mm), and small follicles (SF, diameter <3 mm) in CZ black goats during estrus, and we studied and compared miRNA expression patterns among them. We also assessed whether miRNA expression profiles of OS, LF, and SF were consistent, and searched for miRNAs that might be related to follicular maturation in each tissue, with the aim of providing a reference for further studies on molecular mechanisms underlying follicular development in goats. The key miRNAs discovered in this study might be applied as activators or inhibitors in the future to regulate follicular maturation and ovulation, thus improving ovulation and kidding rate in goats and providing high-fertility goats for the goat industry.

## 2. Materials and Methods

### 2.1. Ethics Statements

All experimental procedures and sample collection methods complied with the Regulation on the Administration of Laboratory Animals (CLI.2.293192, 2017 Revision, State Council, China) and were performed in strict accordance with the Institutional Animal Care and Use Committees of South China Agricultural University (approval no. 2018-P002).

### 2.2. Animals and Sample Preparation

We selected CZ black goats from Wens Foodstuff Group Co., Ltd., as the experimental animals (Guangdong, China). Under the condition of natural light, 10 healthy female goats of the same age (approximately 3.5–4.5 years), who had >3 litters, were given ad libitum access to food and water. After estrus synchronization, we slaughtered them, collected their ovaries, and separated the OS, LF, and SF. The LF and SF that were isolated from follicles were analyzed by small RNA-sequencing (small RNA-seq), as previously described [30]. We separated LF, SF, and OS successively. However, at the time of LF separation, due to the fragility of the follicles, the small and medium (3 mm< diameter <10 mm) follicles around the LF were destroyed, as well as the other adjacent large follicles. Because of the complexity of follicle separation, some of these follicles around the LF were broken up, making them unsuitable for subsequent experiments. At last, the OS was isolated from all 10 goats, LF were isolated from 5 goats (1 to 2 LF/goat), and SF were isolated from 6 goats (8–15 SF/goat).

### 2.3. RNA Extraction and Quality Determination

Total RNA was extracted from OS, LF, and SF using TRIzol (Invitrogen, Carlsbad, CA, USA). The RNA quality was estimated using an Agilent 2100 Bioanalyzer (Agilent Technologies, Palo Alto, CA, USA) and NanoDrop spectrophotometer (ND-2000, Thermo Fisher Scientific, Wilmington, DE, USA). The RNA integrity was evaluated by 1% agarose gel; purified RNA was stored at −80 °C. High quality RNA (quantity >6 μg, concentration ≥200 ng/mL, 1.8< OD_260/280_ <2.2, and RNA integrity number >8.5) was used for preparing the cDNA libraries.

### 2.4. Small RNA Library Construction, Sequencing, and Data Processing

After extraction and purification, we used the TruSeq Small RNA Sample Prep Kit (Illumina, San Diego CA, USA) to extract approximately 2 μg of total RNA from each sample to construct the small RNA libraries. All libraries used for high-throughput sequencing of miRNAs were amplified by PCR, with sequencing connectors and index sections added. Next, 18–36 nt RNAs were purified using a 6% Novex TBE PAGE gel (1.0 mm, 10 wells) and quantified with an Agilent 2100 Bioanalyzer. Single-stranded cDNA templates were used for bridge PCR, and Illumina single-end sequencing was performed on a HiSeq 2500 sequencer (Illumina, San Diego, CA, USA). 

Raw reads were handled by a script consisting of read alignment, index trimming, and read counting. To obtain clean reads, raw reads were further filtered as per the following rules: (1) removing low-quality reads containing a >1 low quality (Q-value ≤ 20) base or unknown nt (N); (2) removing reads without 3′ adapters; (3) removing reads with 5′ adapters; (4) removing reads containing 3′ and 5′adapters but without small RNA fragments between them; (5) removing reads containing poly-A from small RNA fragments; and (6) removing reads that were <18 nt (excluding adapters). The excised clean reads were mapped to the goat reference genome (Capra_hircus.ARS1.DNA.toplevel.fa, Ensembl V95.1) using miRDeep2 (v2.0.0.8), in which the mapper.pl program called Bowtie was used to align the anti-repeat sequence with the reference genome sequence. The de-repeat sequences were aligned in miRBase (http://www.mirbase.org/) with mature and precursor miRNA sequences of the species, and the detected miRNAs were annotated. A novel miRNA prediction analysis was performed based on MIREAP analysis of the unannotated sequences. The read count values of the miRNAs were calculated based on the number of mature miRNA sequences. Herein, we analyzed the expression difference of the known miRNAs.

### 2.5. Identification of Differentially Expressed miRNAs (DEmiRNAs) 

For small RNA-seq analysis, due to the short length of the miRNA, there is no need for length correction in the standardization of the expression level. Instead, counts per million (CPM) values were used for inter-sample correction of total reads (CPM = C/N × 1,000,000; C = total number of reads mapped to the genes, and N = total number of mapped reads). The density distribution of the CPM values was estimated using the density function in R (v4.0.1). Principal components analysis (PCA) was performed by the procmp function in R. Differential expression in each group was identified by DESeq (v1.18.0) with R, and DEmiRNAs were identified by |log2FC| > 1 and *p* < 0.05. 

The ggplots2 package in R was used to generate volcanic plots with the DEmiRNAs, and the heatmap package in R was used to cluster all miRNAs. The Euclidean distance was calculated according to the expression levels of the same miRNA in different samples and the expression patterns of different miRNAs in the same sample. The complete linkage hierarchical clustering method with the largest intercluster distance was used for clustering. 

### 2.6. Target Gene Prediction

Based on possible functional relationships between the DEmiRNAs and genes, a systematic bioinformatic analysis was performed. The target genes of the DEmiRNAs were predicted by miRanda, with the 3′-UTR mRNA sequence serving as the target sequence. The parameters used to predict the miRNA–target interaction were a mapping score >140 and free energy <1.0. Cytoscape (v3.5.1) was then used for visualization of the DEmiRNAs and target genes, and the degree represented the number of target genes of the DEmiRNAs. 

### 2.7. Functional Enrichment Analysis 

The target genes of the DEmiRNAs were analyzed by gene ontology (GO, http://geneontology.org/) and the Kyoto Encyclopedia of Genes and Genomes (KEGG, http://www.kegg.jp/) in DAVID (https://david.ncifcrf.gov). The degree of enrichment was measured by the Rich factor, false discovery rate, and number of genes enriched in a pathway. A *p* ≤ 0.05 indicated statistical significance.

## 3. Results

### 3.1. Quality Control of Sequencing Data

To identify and characterize the miRNAs during the follicular phase in goat ovaries, we constructed small RNA libraries of OS, LF, and SF. Data pertaining to LF and SF were based on a previously published study [30], and these follicles were from uniparous and multiple goats. We have previously published a miRNA study, the findings of which can be potentially related to high fertility, but the miRNA-based study on follicular maturation in CZ black goats has not been published as yet. The small RNA-seq results revealed that the GC content was around 50%, and Q30 was >90%; in addition, after filtering, the percentage of clean reads was between 45.40% and 97.35%, of which 60.45–91.57% were mapped to the goat genome (Table 1, Appendix A for details); these met our analysis requirements and could be further analyzed. We additionally counted the number of clean reads with a length of 18–36 nt and found that the length of the small RNA basically included all types of miRNAs, among which the distribution of miRNAs with a length of 22 nt was the most prominent (Figure 1).

### 3.2. Small RNA Annotation Analysis

All clean reads were annotated and classified by alignment against the Rfam11.0 and GenBank databases (Appendix A). To ensure that each small RNA was uniquely annotated, the annotation results were arranged according to the priority of knowledge: miRNA > piRNA > rRNA > tRNA > snRNA > snoRN > novel miRNA. Before de-duplication, the proportion of known miRNAs and unknown functional RNAs in all libraries was 49–65% and 33–48%, respectively (Figure 2A–C). After de-duplication, the proportion of known miRNAs and unknown functional RNAs in all libraries was 2–3% and 94–96%, respectively (Figure 2D–F). Further, the proportion of known mature miRNAs, precursor miRNAs, and novel miRNAs in all libraries was 24–28%, 15–18%, and 54–61%, respectively (Figure 2G–I). To summarize, unknown RNAs and novel miRNAs were the most dominant, indicating that there were a large number of low expression and unknown miRNA sequences to be identified and studied.

### 3.3. Analysis of DEmiRNAs

Based on the heatmap shown in Figure 3A, most miRNAs were not differentially expressed in OS, LF, and SF, and only a few miRNAs were differentially expressed according to the miRNA expression abundance, indicating that the expression profiling of the miRNAs in OS, LF, and SF was similar. Before the difference analyses, 436, 433, and 435 miRNAs were found in OS vs. LF, OS vs. SF, and LF vs. SF. Upon comparing OS vs. LF, OS vs. SF, and LF vs. SF, 32, 16, and 29 DEmiRNAs were identified, respectively (Appendix A). Three DEmiRNAs were common in the three group (Figure 3B). The top 10 DEmiRNAs with a significant difference (*p* < 0.05) are listed in Table 2. The PCA plot is shown in Appendix A.

### 3.4. Prediction and Analysis of Target Genes of DEmiRNAs

We herein used miRanda to predict the target genes of the DEmiRNAs (Appendix A). In OS vs. LF, there were 4541 predicted targets for 32 DEmiRNAs and 20,730 targeted relationships (Figure 4A), and the top 5 significant miRNAs were miR-543-5p (degree = 1026), miR-543-3p (degree = 1019), miR-449b-5p (degree = 1009), miR-143-3p (degree = 1003), and miR-656 (degree = 996). In OS vs. SF, there were 3869 predicted targets for 16 DEmiRNAs and 9927 targeted relationships (Figure 4B), and the top 5 significant miRNAs were miR-412-3p (degree = 919), miR-1185-3p (degree = 898), miR-2404 (degree = 890), miR-182 (degree = 836), and miR-206 (degree = 716). In LF vs. SF, there were 4447 predicted targets for 29 DEmiRNAs and 19,732 targeted relationships (Figure 4C), and the top 5 significant miRNAs were miR-195-5p (degree = 1469), miR-502b-5p (degree = 1341), miR-495-3p (degree = 1236), miR-494 (degree = 1069), and miR-1248-5p (degree = 1023).

Most miRNAs were predicted to regulate hundreds or even thousands of target genes at the same time. In OS vs. LF, OS vs. SF, and LF vs. SF, it was predicted that the DEmiRNA with the most target genes was miR-543-5p, miR-412-3p, and miR-195-5p, respectively, and the number of target genes was 1026, 919, and 1469, respectively. Many target genes may be regulated by multiple miRNAs. In OS vs. LF, LIM and calponin-homology domains 1 (*LIMCH1*) was predicted to be targeted by 17 DEmiRNAs. In OS vs. SF, both zinc finger protein 827 (*ZNF827*) and myocardin-related transcription factor B (*MRTFB*) were predicted to target 11 DEmiRNAs. Finally, in LF vs. SF, it was predicted that the new gene *ENCHIG000015853* was targeted by 19 DEmiRNAs. 

### 3.5. Functional Analysis of Target Genes

GO functional enrichment analyses (Appendix A) revealed that 1086 GO terms were significantly enriched in OS vs. LF (*p* < 0.05), such as tissue development, extracellular space, and sequence-specific DNA binding of the transcription regulatory region. In OS vs. SF, 1132 GO terms were significantly enriched (*p* < 0.05), such as tissue development, cell differentiation regulation, and GTPase regulator activity. In LF vs. SF, 1032 GO terms were significantly enriched (*p* < 0.05), such as cell proliferation, neuron synapse, and glycosaminoglycan binding. The top 10 GO terms in each category with a significant difference (*p* < 0.05) are shown in Figure 5A–C. Further, comprehensive comparison showed that there were 612 GO terms in the triple comparison (OS vs. LF, OS vs. SF, and LF vs. SF); 87 terms were present in OS vs. LF and OS vs. SF; 127 terms in OS vs. LF and LF vs. SF; and 63 terms in OS vs. SF and LF vs. SF; moreover, 260, 217, and 83 terms were only significantly enriched in OS vs. LF, OS vs. SF, and LF vs. SF, respectively (Figure 5D).

Based on the KEGG pathway enrichment analysis (Appendix A), in OS vs. LF, there were 39 signaling pathways (*p* < 0.05) enriched by the target genes of the DEmiRNAs, including the TGF-β signaling pathway, cell adhesion molecule, MAPK signaling pathway, and oocyte meiosis. In OS vs. SF, there were 31 signaling pathways (*p* < 0.05) enriched by the target genes of the DEmiRNAs, including the cell adhesion molecule, TGF-β signaling pathway, oocyte meiosis, Notch signaling pathway, and ovarian steroid production. In LF vs. SF, there were 41 signaling pathways (*p* < 0.05) enriched by the target genes of the DEmiRNAs, including the cell adhesion molecule, TGF-β signaling pathway, cell cycle, and MAPK signaling pathway. The top 20 significantly enriched (*p* < 0.05) pathways are shown in Figure 6A–C. 

A comprehensive comparison showed that 19 signaling pathways were enriched in the triple comparison (OS vs. LF, OS vs. SF, and LF vs. SF); 2 signaling pathways were enriched in OS vs. LF and OS vs. SF; 4 in OS vs. LF and LF vs. SF; and 4 in OS vs. SF and LF vs. SF. Further, 14, 6, and 14 signaling pathways were only significantly enriched in OS vs. LF, OS vs. SF, and LF vs. SF, respectively (Figure 6D).

## 4. Discussion

The ovary is an important reproductive organ that directly affects the estrous cycle and reproductive ability in mammals [31]. Many studies have been conducted on follicles at different stages of development [14,32], but few exist on OS, and thus, regulatory differences between OS and follicles remain unclear. Goats, as a single or twin [4], represent a good animal model to study the mechanisms related to follicular development. To investigate the miRNA expression profile of OS, LF, and SF in CZ black goats and to reveal the genetic differences among them at the transcriptome level, we, for the first time, used high-throughput sequencing. According to the literature reports, gene expression patterns can be judged according to the difference in expression abundance [33,34,35]. In this study, the obtained results showing that most miRNAs were not differentially expressed in OS, LF, and SF, only a few miRNAs were differentially expressed according to the miRNA expression abundance (Figure 3 in the results). Therefore, we speculated that the miRNAs expression patterns of OS, LF and SF were similar, suggesting that their main functions are similar, which include maintaining follicular development and maturation. In terms of follicular maturation, it is of great significance to study the role of specific miRNAs in OS, LF, and SF. The findings of this study provide a theoretical basis to further elucidate the role of miRNAs in follicular maturation.

Overall, there were 16 miRNAs with CPM > 10,000 in at least one tissue (OS, LF, or SF), among which 2 miRNAs (miR-143-3p and miR-21-5p) were significantly different in OS vs. LF. In a study on ovarian cancer, miR-143-3p was noted to be potentially serve as a tumor suppressor, and its expression upregulated in normal ovarian tissue [36]. Further, miR-143-3p has been reported to inhibit the proliferation, migration, and invasion of ovarian cancer cells in vitro, and also inhibit the occurrence of ovarian cancer in vivo [37]. Another study found that the expression of miR-143-3p was the highest (expression level: OS > SF > LF) and that it may play an important role in the development of OS. Moreover, miR-21-5p was found to be highly expressed in atretic follicles and early corpus luteum [38,39]. It has also been reported to affect the development potential of mouse and bovine oocytes [40], as well as regulate high fecundity [41]. The study indicated that the expression level of miR-21-5p was in the order of OS > SF > LF; it was only differentially expressed in OS vs. LF and may be involved in the functional regulation of OS.

We then screened miRNAs that were only expressed in one tissue. The miRNAs in at least half of the samples with CPM ≥ 1 were considered as expressed and those with CPM < 1 were considered as not expressed. Accordingly, only 8, 21, and 5 miRNAs were expressed in OS, LF, and SF, respectively. Only 7 miRNAs (miR-187, miR-146b-3p, miR-154b-3p, miR-411b-3p, miR-656, miR-2332, and miR-502b-5p) were significantly differentially expressed, and only miR-187 was found to be related to ovarian function. miR-187 is involved in the regulation of ovarian cancer by targeting disabled homolog-2, which supposedly promotes tumor progression in advanced cancers via epithelial–mesenchymal transition [42,43]. In addition, miR-187 has been found to be significantly upregulated in OS compared with LF and it might target *INHBB*, which plays a key role in the apoptosis of granulosa cells and regulation of the cell cycle [44], and is also vital for early follicular development in mice [45]. Moreover, miR-187 seems to be involved in regulating the development of OS. It has been found that miR-430c, miR-26a, and miR-202-5p are gonadal specific or sex biased in gonadal development, which might be a crucial candidate in sex differentiation and development [46]. Therefore, these key miRNAs identified in our study might be crucial candidate in OS, LF, and SF during ovarian maturation.

To further screen reliable DEmiRNAs, three screening conditions (CPM ≥ 1, |log_2_(FC) |> 1, and *p* < 0.05) were used, and 15, 25, and 6 DEmiRNAs were found in OS, LF, and SF, respectively. Specifically, miR-182, miR-122, and miR-206 were significantly upregulated in OS compared with the other two tissues. miR-182, miR-122, and miR-206 have been reported to regulate oocyte maturation and granulosa cell development [47,48,49]; they evidently regulate oocyte meiosis by targeting *SMC3*, *SGO1,* and *AURKA*, respectively. *SMC3* promotes oocyte apoptosis and affects the pathogenesis of polycystic ovary syndrome [50,51]. *AURKA* is a rate-limiting factor that promotes microtubule growth as oocytes resume meiosis [52]. *SGO1* plays a key role in the centromere cohesion of sister chromatids and movement of chromosomes to the spindle pole [53], and its downregulation has been shown to reduce the speed and quality of embryo development [54]. Hence, these miRNAs may play a role in ovarian organogenesis and hormone secretion through oocyte meiosis. Besides, miR-202-5p and miR-485-3p were significantly upregulated in LF; they have been indicated to have a regulatory role in ovarian hormone metabolism and granulosa cell development [55,56]. In zebrafish, knocking out miR-202 impaired the early steps of oogenesis/folliculogenesis and decreased the number of large (i.e., vitellogenic) follicles, ultimately leading to no reproductive success [57]. In this study, miR-202-5p and miR-485-3p were noted to be involved in the MAPK signaling pathway by targeting *EREG* and *FLT3*, respectively. *EREG* can be induced by gonadotropin-releasing hormone and regulates ovulation [58,59]. *FLT3* is highly expressed in oocytes and related to follicular growth and maturation [60]; moreover, it has been reported to be highly expressed in ovarian cancer tissues, promoting ovarian cancer metastasis and angiogenesis [61]. Thus, it appears that miR-202-5p and miR-485-3p play a role in follicular maturation by targeting *EREG* and *FLT3*, respectively. In addition, miR-2404 was significantly upregulated in SF compared with OS, but there was no significant difference compared with LF. It is noteworthy that miR-2404 has not been found to be related to ovarian development. Through target gene prediction of miRNAs, *BMP7* and *CDKN1C* targeted by miR-2404 were found to participate in the TGF-β signaling pathway and cell cycle, respectively. *BMP7* is an active regulator of the transformation from primordial to primary follicle [62], promoting the expression of *FSHR* in human granulosa cells [63] and potentially serving as a key factor in the development of ewe follicles [64]. *CDKN1C* is highly expressed in mouse ovaries [65] and it may be an imprinted gene in oocytes [66,67]. Accordingly, miR-2404 might play a regulatory role in follicular development by targeting *BMP7* and *CDKN1C*.

Finally, in the triple comparison (OS vs. LF, OS vs. SF, and LF vs. SF), the expression of miR-1185-3p, miR-190b, and miR-487a-3p was significantly different (expression levels: LF > SF > OS). The functions of these three have not been reported as yet; we herein speculate their possible functions by analyzing the functions of their target genes. We found that the Hippo signaling pathway, cell adhesion molecules, and TGF-β signal pathway were the key elements that may be involved in the ovarian function and follicular maturation. As the potential role of these miRNAs in ovarian maturation was predicted, further experiments were needed to verify their function. Specifically, as reported in the literature [56], a series of functional experiments must be taken to identify their location and their upstream/downstream regulators, in order to reveal the mechanisms of ovarian maturation.

The Hippo signaling pathway has a key role in regulating the proliferation and apoptosis of mammalian cells and in maintaining ovarian function as well as promoting follicular growth [68,69]. *AREG* [70,71], *WWTR1* [72,73], and *CTGF* [68,74] are important genes in this pathway, which are involved in the regulation of follicular development, atresia, ovulation, and luteal function in mammals. Herein *AREG*, *WWTR1,* and *CTGF* were predicted to be the target genes of miR-1185-3p, suggesting that miR-1185-3p regulates them through the Hippo signaling pathway, in turn regulating ovarian function and follicular development. Cell adhesion molecules are essential for the separation and aggregation of different cell types to form different tissues; they maintain the integrity of the germ cells in female gonads [75,76]. *CLDN10* [77,78] and *CADM1* [79] in this pathway play an important role in regulating the function of OS and development of cumulus cells. In this study, we found that miR-1185-3p targeted *CLDN10* and *CADM1* and mediated the cell adhesion molecules to regulate ovarian function. Further, the TGF-β signaling pathway is widely involved in functions such as cell proliferation, apoptosis, differentiation, and migration, which affect follicular development, embryonic development, and organ formation [80,81]. *BMP7* [82], *BMP5* [83], *FMOD* [84], *TGFB2* [85], *BMPR1B* [86], and *SMAD7* [87] are the main genes in this pathway, and they are closely related to granulosa cell development, follicular maturation, and ovulation. We noted that miR-1185-3p targeted *BMP7*, *BMP5*, and *FMOD*; miR-190b targeted *TGFB2* and *BMPR1b*; and miR-487a-3p targeted *TGFB2* and *SMAD7*; thus, they seem to mediate the TGF-β signaling pathway to regulate the ovulation rate.

## 5. Conclusions

Overall, based on the expression profiling of miRNAs, we speculated that the miRNA expression patterns of OS, LF and SF were similar. Most miRNAs were not differentially expressed in OS, LF, and SF; only a few miRNAs were differentially expressed according to the miRNA expression abundance. Our data suggested that miR-182 (*SMC3*), miR-122 (*SGO1*), and miR-206 (*AURKA*) were involved in ovarian organogenesis and hormone secretion by oocyte meiosis, whereas miR-202-5p (*EREG*) and miR-485-3p (*FLT3*) were involved in follicular maturation through the MAPK signaling pathway. Moreover, miR-2404 (*BMP7* and *CDKN1C*) was observed to play a key role in follicular development through the TGF-β signaling pathway and cell cycle. In addition, the expression of miR-1185-3p, miR-190b, and miR-487a-3p may increase with follicular development and maturation, and they seemed to play a pivotal role through the Hippo signaling pathway, cell adhesion molecules, and TGF-β signaling pathway. In a future study, we aim to verify the relationship between candidate miRNAs and their target genes and study the mechanism of their influence on follicular maturation so as to reveal the mechanisms underlying miRNAs in goat ovarian follicular maturation. 

## Figures and Tables

**Figure 1 genes-11-00970-f001:**
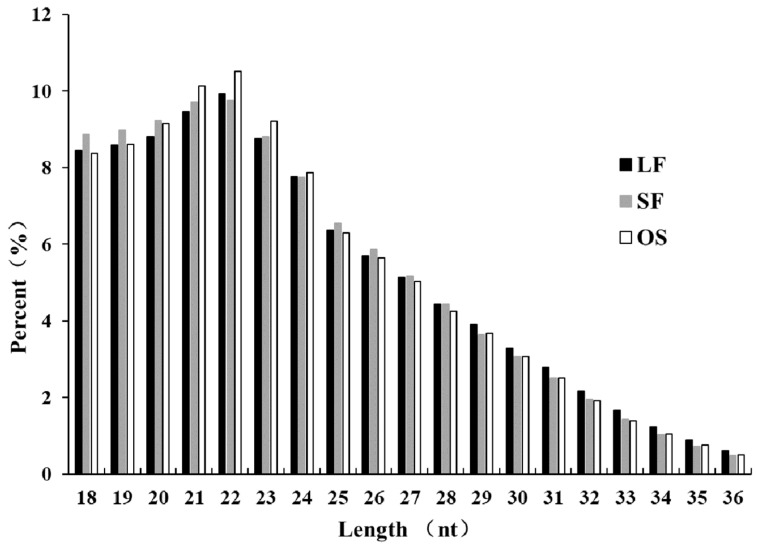
Length distribution of the small RNAs after de-duplication. OS: ovarian stroma; LF: large follicles; SF: small follicles.

**Figure 2 genes-11-00970-f002:**
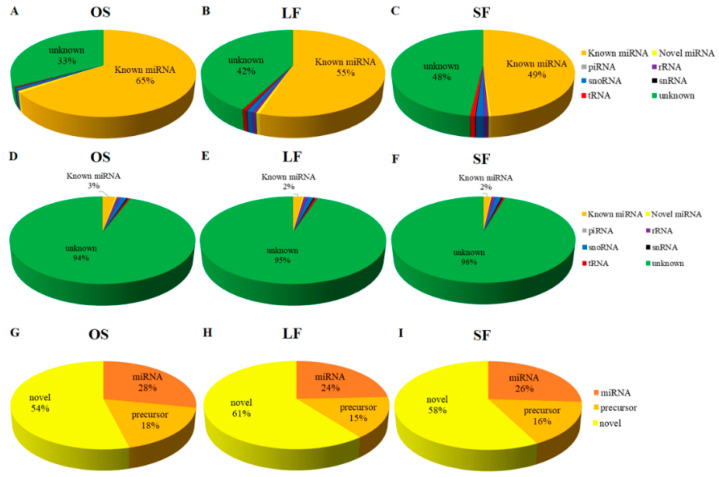
Annotation classification of the small RNAs. (**A**–**C**) Sequence proportion of OS, LF, and SF annotated to various small RNAs before de-duplication and (**D**–**F**) after de-duplication. (**G**–**I**). Ratio of small RNA annotation to mature miRNAs, miRNA precursors, and new miRNAs in OS, LF, and SF.

**Figure 3 genes-11-00970-f003:**
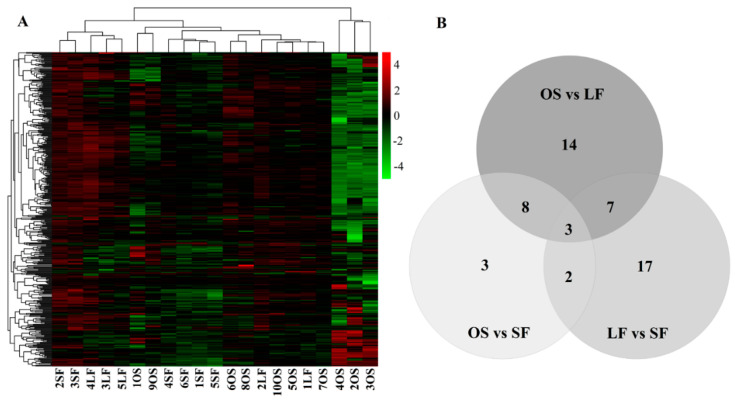
Expression profiling of miRNAs in OS vs. LF, OS vs. SF, and LF vs. SF. (**A**) Heatmap of all miRNAs. (**B**) Venn map of DEmiRNAs in OS vs. LF, OS vs. SF, and LF vs. SF.

**Figure 4 genes-11-00970-f004:**
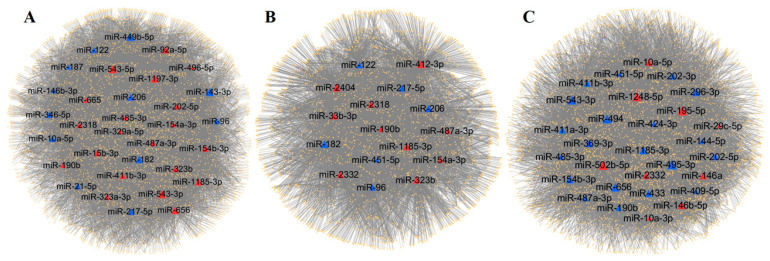
Target genes of the DEmiRNAs. (**A**) Interaction network of the DEmiRNAs–target genes in OS vs. LF, (**B**) in OS vs. SF, and (**C**) in LF vs. SF.

**Figure 5 genes-11-00970-f005:**
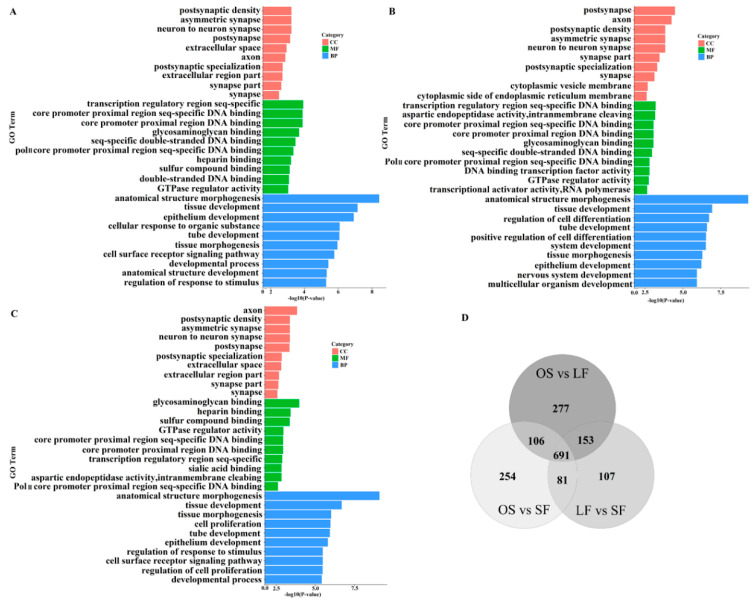
GO functional enrichment analyses of the DEmiRNAs. (**A**) Top 10 differential GO terms of target genes from each category in OS vs. LF, (**B**) in OS vs. SF, and (**C**) in LF vs. SF. (**D**) Venn diagram of GO terms in OS vs. LF, OS vs. SF, and LF vs. SF.

**Figure 6 genes-11-00970-f006:**
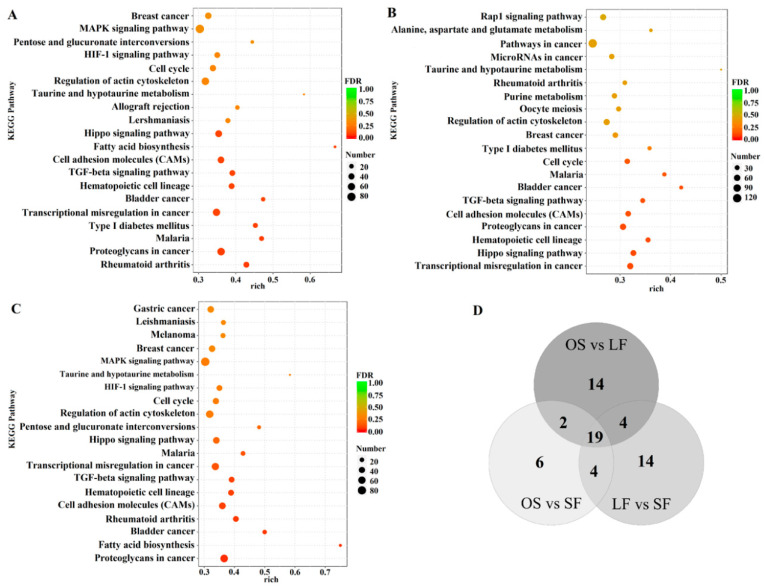
Top 20 significantly enriched KEGG pathways (**A**) in OS vs. LF, (**B**) in OS vs. SF, and (**C**) in LF vs. SF. (**D**) Venn diagram of the KEGG pathways in OS vs. LF, OS vs. SF, and LF vs. SF.

**Table 1 genes-11-00970-t001:** Quality control of the small RNA-seq data.

Sample	GC (%)	Q30 (%)	Clean Ratio (%)	Mapped Ratio (%)	Sample	GC (%)	Q30 (%)	Clean Reads (%)	Mapped Ratio (%)
1OS	49.46	96.74	97.35	91.57	1LF	49.95	96.66	92.40	86.39
2OS	49.56	92.36	53.20	60.45	2LF	50.48	94.68	84.74	82.44
3OS	49.56	91.99	45.40	64.67	3LF	51.78	95.03	73.61	76.28
4OS	49.56	91.41	52.73	67.66	4LF	51.79	94.57	77.25	71.45
5OS	50.30	94.77	86.42	84.08	5LF	51.16	95.01	80.09	73.91
6OS	50.61	94.63	81.29	82.93	1SF	50.80	96.48	88.44	77.63
7OS	51.12	96.50	82.04	75.03	2SF	50.72	94.29	82.43	78.71
8OS	50.27	95.35	86.29	84.22	3SF	51.98	93.69	60.17	79.80
9OS	50.38	94.79	86.30	86.33	4SF	51.33	94.48	84.46	72.34
10OS	50.42	94.99	83.79	86.20	5SF	50.76	95.17	83.72	75.22
					6SF	51.29	94.96	75.81	72.53

**Table 2 genes-11-00970-t002:** Top 10 DEmiRNAs with a significant difference.

miRNA	log_2_(FC)	*p*-Value	miRNA	log_2_(FC)	*p*-Value
**OS vs. LF**			**OS vs. SF**		
miR-190b	3.85	1.04 × 10^−4^	miR-2318	3.06	9.61 × 10^−6^
miR-487a-3p	2.70	2.97 × 10^−4^	miR-190b	2.20	4.87 × 10^−3^
miR-1185-3p	2.72	7.04 × 10^−4^	miR-122	−9.76	5.02 × 10^−3^
miR-543-5p	2.79	1.28 × 10^−3^	miR-182	−10.23	6.38 × 10^−3^
miR-154b-3p	2.61	2.16 × 10^−3^	miR-412-3p	2.24	6.97 × 10^−3^
miR-2318	2.57	2.22 × 10^−3^	miR-2332	2.17	7.10 × 10^−3^
miR-154a-3p	2.25	2.82 × 10^−3^	miR-206	−9.14	2.00 × 10^−2^
miR-656	1.54	3.43 × 10^−3^	miR-487a-3p	1.37	2.18 × 10^−2^
miR-411b-3p	1.91	3.44 × 10^−3^	miR-96	−10.53	2.23 × 10^−2^
miR-323b	2.64	7.12 × 10^−3^	miR-323b	1.46	3.37 × 10^−2^
**LF vs. SF**					
miR-424-3p	−1.35	2.23 × 10^−4^	miR-195-5p	1.05	5.47 × 10^−3^
miR-202-3p	−1.59	1.09 × 10^−3^	miR-146b-5p	1.30	5.50 × 10^−3^
miR-10a-3p	1.54	1.60 × 10^−3^	miR-190b	−1.61	7.14 × 10^−3^
miR-1248-5p	1.30	3.56 × 10^−3^	miR-411a-3p	−1.21	7.16 × 10^−3^
miR-409-5p	−1.24	3.72 × 10^−3^	miR-202-5p	−1.31	7.80 × 10^−3^

## Data Availability

Our sequencing data could be found in the sequence read archive (SRA) of NCBI. The BioProject ID of follicles was PRJNA579007 and PRJNA579194. In addition, one of our ongoing studies also includes the data on ovarian stroma, which will be available soon at the time of publication. If any researchers need this data, please contact us by email. After signing the relevant agreement, we will share the data.

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
