# Peer review of "A Preliminary Study on the Characteristics of microRNAs in Ovarian Stroma and Follicles of Chuanzhong Black Goat during Estrus"

_genes, 2020, doi:10.3390/genes11090970_

Round 1

Reviewer 1 Report

The literature review, justification of the study(application of the study findings) is scanty.

The goats being studied are described as excellent in production rate-ln 62-65-why study these goats?.

Many statements are reported in the past tense-needs English editing.

How the study results are going to help the goat industry to increase productivity and the economic benefit is not explained both in the background and the conclusion.

Reviewer 2 Report

Comments

There manuscript address important in goat ovarian follicles and stroma during estrus using microRNA techniques. The paper overall needs some major grammatical changes to help improve the clarity of the major results and conclusions. I have provided some suggested changes below:

Abstract

Line 25 change participated in ….. to associated or were involved…

Line 31 and 32 How will it help improve it? will help to improve 31 ovulation rate and kidding rate in goat.

Introduction

 Lien 36 and 37 write in present tense ….change was to is.  Fecundity was a complex quantitative character, which was mainly restricted by genetic factors 36 and affected the economic effect of goat industry.

Line 41  Change contained to contain.   As a dynamic reproductive organ, ovary contained ovarian

Line 44 and 45 Be specific, it refers to what? Therefore, in the study of ovary, it is necessary to subdivide it ……..

Line 46 change was to is…. MicroRNA (miRNA) was a kind…….

Line 48 change was to is …miRNA was mainly

Line 58 change was to is …Chuanzhong (CZ) black goat (Capra hircus) was an

Line 53-58 Need to provide more details of  why its important to study the MiRNA in goat. Has there been previous miRNA studies in goat, but not in the follicles? Highlight that.

Materials and methods

Line 86-87 Why the justification for the variation in sample size for the OS, LF and SF? Because the n=10 goats. Explain. OS was isolated from 10 goats; LF 86 were isolated from 5 goats, with 1-2 LF per goat; SF were isolated from 6 goats, with 8-15 SF per goat.

Results

Line 149 Change were shown to showed that …..The statistical results of small RNA-seq were shown that…..

Line 181-182 How can they be similar and mixed together at the ssame time? Need to explain clearly.

Based on the heatmap (Figure 3A), we found that the expression profiling of miRNAs in OS, LF 182 and SF were similar, and they were mixed together

Line 187..change were to are… were shown in Table 2.

Discussion

Line 257 change was to is….Ovary was an important reproductive organ

Line 259 Change sentence to ..Currently, there are several studies…..At present, there were more and more studies

Line 328-332 Since the function of these i.e. miR-1185-3p, miR-190b and miR-487a-3p are not yet defined, it is important to along side the computational speculation, further biological research will be required. Explain it and suggestion it a possible future direction of research to be done.

Conclusion

Need to state the overall conclusions clearly. How were similar were the miRNA expression patterns of OS, LF and SF? What is the biological significance of key miRNAs expressed?

Round 2

Reviewer 1 Report

72 CZ black goat plays an important role in animal husbandry [25-27]. how does a goat play a role in animal husbandry?
76 sequence the transcriptome of ovarian stroma (Did you sequence the whole transcriptome or only miRNA? )
79 We also assessed whether miRNA expression profiles of OS, LF, and SF were consistent at the
transcriptome level (the meaning of this statement is not clear)

98 At the time of LF separation, the follicles around LF were sacrificed, including SF and mid-follicles (3 mm < d < 10 mm), even LF nearby (this statement is unclear)

198-200 Upon comparing OS vs. LF, OS vs. SF, and LF vs. SF, 436, 433, and 435 miRNAs were identified a and only 32, 16, and 29 of them were differentially expressed, respectively (Table S3); of them, only 8, 7, and 2 DEmiRNAs were differentially expressed, respectively (statement unclear –needs rewriting)

280 and SF, only a few miRNAs were differentially expressed according to the miRNA expression
abundance (Figure 3).(referencing figures in discussion section?)

374 we speculate that the miRNA expression patterns of OS, LF and SF were similar (is this a speculation or the study findings?)
